# Structure determination of Murine Norovirus NS6 proteases with C-terminal extensions designed to probe protease–substrate interactions

Humberto Fernandes*, Eoin N. Leen, Hamlet Cromwell Jr, Marc-Philipp Pfeil** and Stephen Curry

Department of Life Sciences, Imperial College London, UK
* Current affiliation: Institute of Biochemistry and Biophysics, Polish Academy of Sciences, Warsaw, Poland
** Current affiliation: Department of Biochemistry, University of Oxford, Oxford, UK

Corresponding author
Humberto Fernandes,
hfernandes@ibb.waw.pl

## ABSTRACT

Noroviruses are positive-sense single-stranded RNA viruses. They encode an NS6 protease that cleaves a viral polyprotein at specific sites to produce mature viral proteins. In an earlier study we obtained crystals of murine norovirus (MNV) NS6 protease in which crystal contacts were mediated by specific insertion of the C-terminus of one protein (which contains residues P5-P1 of the NS6-7 cleavage junction) into the peptide binding site of an adjacent molecule, forming an adventitious protease-product complex. We sought to reproduce this crystal form to investigate protease–substrate complexes by extending the C-terminus of NS6 construct to include residues on the C-terminal (P′) side of the cleavage junction. We report the crystallization and crystal structure determination of inactive mutants of murine norovirus NS6 protease with C-terminal extensions of one, two and four residues from the N-terminus of the adjacent NS7 protein (NS6 1′, NS6 2′, NS6 4′). We also determined the structure of a chimeric extended NS6 protease in which the P4-P4′ sequence of the NS6-7 cleavage site was replaced with the corresponding sequence from the NS2-3 cleavage junction (NS6 4′ 2|3). The constructs NS6 1′ and NS6 2′ yielded crystals that diffracted anisotropically. We found that, although the uncorrected data could be phased by molecular replacement, refinement of the structures stalled unless the data were ellipsoidally truncated and corrected with anisotropic $B$-factors. These corrections significantly improved phasing by molecular replacement and subsequent refinement. The refined structures of all four extended NS6 proteases are very similar in structure to the mature MNV NS6—and in one case reveal additional details of a surface loop. Although the packing arrangement observed showed some similarities to those observed in the adventitious protease-product crystals reported previously, in no case were specific protease–substrate interactions observed.

# INTRODUCTION

Noroviruses are responsible for over half of the outbreaks of gastroenteritis worldwide (*Karst, 2010*). They belong to the *Caliciviridae*, a family of positive-sense, single-stranded RNA viruses with a ~7.5 kb genome that generally contains three open reading frames (ORF) (*Jiang et al., 1993*; *Lambden et al., 1993*; *Glass et al., 2000*); a novel fourth ORF was recently identified in Murine norovirus (MNV) (*McFadden et al., 2011*). Whereas ORF2, ORF3 and ORF4 each encode single proteins, ORF1 codes for a large polyprotein (~190 kDa) that is processed by the virally-encoded protease at five specific sites to release the six 'mature' non-structural proteins (NS1/2–NS7)—and an array of functional precursors—that are required for virus replication (*Belliot et al., 2003*; *Sosnovtsev et al., 2006*; *Muhaxhiri et al., 2013*). The viral NS6 protease present within the C-terminal half of the polyprotein performs all the processing, including its own autocatalytic release from the precursor (*Liu, Clarke & Lambden, 1996*; *Belliot et al., 2003*; *Sosnovtsev et al., 2006*; *Scheffler et al., 2007*).

Crystal structures have been determined for the NS6 proteases from several norovirus strains (Chiba virus, Murine norovirus, Norwalk virus, Southampton norovirus) (*Nakamura et al., 2005*; *Zeitler, Estes & Prasad, 2006*; *Hussey et al., 2011*; *Kim et al., 2012*; *Leen, Baeza & Curry, 2012*; *Muhaxhiri et al., 2013*). The solution structure and dynamics of Norwalk virus NS6 have also been analysed by NMR spectroscopy (*Takahashi et al., 2013*). Norovirus NS6 is a cysteine protease with a chymotrypsin-like fold: two $\beta$-barrel domains separated by a cleft that contains a Cys-His-Asp/Glu catalytic triad similar in arrangement to the Ser-His-Asp triad characteristic of serine proteases (*Allaire et al., 1994*; *Matthews et al., 1994*). Calicivirus NS6 is related in sequence and structure to the picornavirus 3C proteases, which have the same role in polyprotein processing for these single-stranded RNA viruses (*Leen, Baeza & Curry, 2012*).

Previously our lab determined the structure of full-length Murine norovirus NS6 (residues 1-183) (*Leen, Baeza & Curry, 2012*). Strikingly, adventitious crystal contacts placed the C-terminus of one molecule in the active site of another, thereby generating the structure of a protease-product complex that plausibly represents the final step of the *trans* cleavage by NS6 at the NS6–NS7 junction. An almost identical packing arrangement was obtained in a different space-group with crystals of Norwalk virus NS6 protease (*Muhaxhiri et al., 2013*). These structures, together with the structures of di-, tri- and penta-peptidyl substrate analogues bound to human norovirus NS6 structures (*Hussey et al., 2011*; *Kim et al., 2012*; *Muhaxhiri et al., 2013*) revealed details of the specific contacts made by the P5-P1 residues of peptidyl products of the protease (Fig. 1); in particular they showed the anchoring of P1-Gln by specific H-bond interactions with Thr 134 and His 157 (Fig. 1A), and the accommodation of the hydrophobic side-chains of P2–Phe and P4-Leu in apolar pockets (Fig. 1B). However, to date there are no structural data on protease–substrate complexes that might uncover details of the interactions made by the protease with the C-terminal, prime side of noroviral peptide cleavage junctions (residues P1′-P4′).

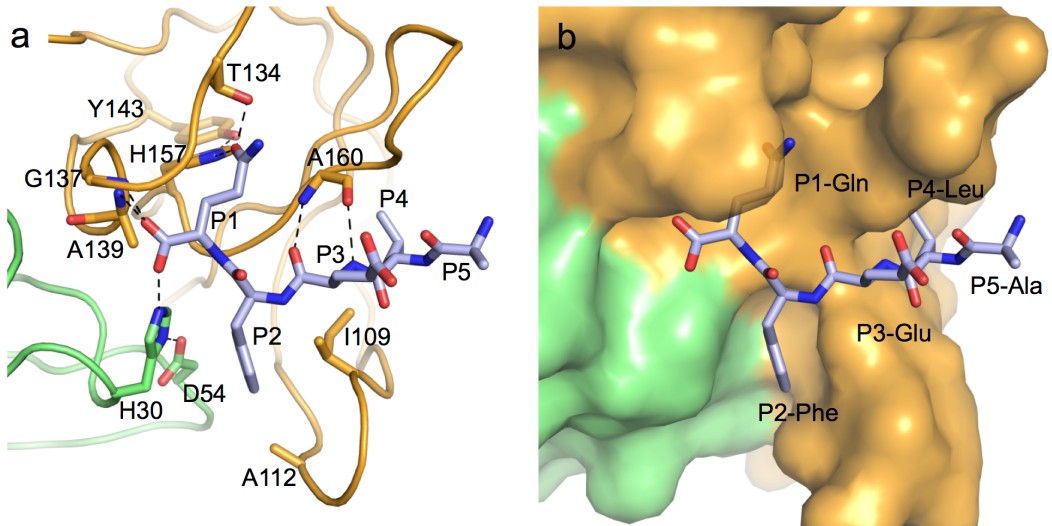

**Figure 1** **Specific protease-product interactions observed in the original crystals of MNV NS6$^{pro}$.** (A) The N- and C-terminal domains of one protease molecule are coloured green and orange respectively. The C-terminus of an adjacent protease that is accommodated specifically in the substrate-binding site (residues P1-P5) is shown as a stick model with grey carbon atoms. Hydrogen-bonds are indicated as dashed lines. (B) Same view as in (A) but with the protease surface shown to illustrate the binding pockets involved in substrate recognition. This figure is a modified version of Fig. 3 from *Leen, Baeza & Curry (2012)* which was published under a Creative Commons CC-BY license. All structural figures were made with PyMOL (*Schrodinger LLC , 2010*).

We reasoned that catalytically inactive MNV NS6 constructs extended at the C-terminus might be able to crystallise with the same packing contacts as we had observed for the full-length protein, and that this could therefore give us a convenient way to investigate the structures of NS6-substrate complexes. To this end we extended an inactive MNV NS6 construct (which incorporates a C139A mutation to knock out the active site nucleophile) by adding residues from NS7, which is immediately downstream in the polyprotein. We made three constructs, NS6 1′, NS6 2′ and NS6 4′, which were extended by 1, 2 and 4 residues respectively, generating proteases that contain substrates that correspond to P4–P1′, P4-P2′ and P4-P4′ of the NS6–NS7 cleavage junction. To investigate the structural variation in substrate recognition we also made an NS6 chimera which interchanged the residues P4-P4′ of the NS6-7 cleavage junction with the sequence of the NS2-3 junction from MNV (NS6 4′ 2|3).

Proteins expressed from all four constructs were crystallised and their structures determined at high resolution by X-ray crystallography. The diffraction patterns from crystals of NS6 1′ and NS6 2′ exhibited marked anisotropy, which stalled the crystallographic refinement at high *R*-factors. However, this was overcome by the successful application of an anisotropic data correction procedure (*Sawaya, 2014*).

The modified MNV NS6 constructs all crystallised with packing arrangements that were distinct from that observed in crystals we previously obtained with the mature NS6 protease—the final 183-residue protein released after processing of the viral polyprotein (*Leen, Baeza & Curry, 2012*; *Muhaxhiri et al., 2013*). Unfortunately, in no case did the

packing arrangements involve insertion of the cleavage junction in the extended C-terminus into the active site of an adjacent protein in a way that allowed the formation of specific protease–substrate interactions. The four new structures reported here therefore do not provide any new information on the mode of binding of the P1′-P4′ residues of junctions cleaved by MNV NS6. However they do confirm the MNV NS6 protease structure and, in at least one case, reveal the structure for the loop connecting β-strands cII and dII that was disordered in the previously reported structure (*Leen, Baeza & Curry, 2012*).

## MATERIALS AND METHODS

### Cloning and purification of MNV NS6 variants

C-terminally extended constructs of the murine norovirus NS6 protease (UniProt accession no. Q80J95; residues 995-1177), inactivated by mutation of the active site Cys 139 to Ala (C139A) were generated by the polymerase chain reaction using the inactive full length MNV NS6 as a template (*Leen, Baeza & Curry, 2012*). The same forward primer was used throughout (5′-CATCATGGATCCGCCCCAGTCTCCATCTGG). To create constructs extended by one (Gly; NS6 1′), two (Gly-Pro; NS6 2′) or four residues (Gly-Pro-Pro-Met; NS6 4′) beyond the natural C-terminus of NS6, we used the reverse primers, 5′-ATGATGAAGCTTAGCCCTGGAACTCCAGAGCCTCAA, 5′-CATCATAAGCTTACGGGCCCTGGAACTCCAGAGCCTCAAGTGTGGGTTCT-CCGTGAGT and 5′-CATCATAAGCTTACATCGGCGGGCCCTGGAACTCCAG-AGCCTCAAGTG respectively. To generate the chimera with the P4-P4′ cleavage site of NS2-3 (NS6 4′ 2|3) the reverse primer 5′-TACTACAAGCTTAATCAAACGGG-CCTTCCGCCTGCCAAGCCTCAAGTGTGGGTTCTCCGTGAGT was used. PCR products were digested with BamHI and HindIII and ligated into the pETM-11 vector as described previously for full-length NS6 (*Leen, Baeza & Curry, 2012*). The expressed MNV NS6 variants thus contain a thrombin-cleavable N-terminal His$_6$ tag; processing by thrombin leaves a Gly-Ser di-peptide preceding the Ala1 residue at the N-terminus of our constructs. All plasmid insert sequences were confirmed by DNA sequencing (Eurofins MWG Operon). NS6 1′ and NS6 2′ constructs were transformed into *E. coli* BL21-CodonPlus (Stratagene) and grown in lysogeny broth (LB) supplemented with 25 µg/mL kanamycin and 35 µg/mL chloramphenicol, while NS6 4′ and NS6 4′ 2|3 constructs were transformed into *E. coli* BL21(DE3) pLysS (Promega) and grown in LB supplemented with 25 µg/mL kanmycin and 35 µg/mL chloramphenicol. For large-scale protein expression, 1 L of LB was inoculated with overnight cultures. The cultures were incubated at 37 °C with shaking at 220 RPM and protein expression was induced for 3 to 4 h by addition of 1 mM isopropyl β-D-1-thiogalactopyranoside (IPTG) once the cultures had attained an OD at 600 nm of 1.0. Cells were harvested by centrifugation at 4,000 g for 10 min, and pellets frozen at −80 °C.

*E. coli* pellets containing over-expressed NS6 1′ and NS6 2′ were re-suspended in lysis buffer (50 mM Tris pH 8.0, 300 mM NaCl, 1 mM dithiothreitol (DTT)) supplemented with 0.5 mM PMSF, 2 mg/mL lysozyme, and 0.1% (v/v) Triton X-100 and the cells disrupted by sonication. Cell debris was removed by centrifugation of the bacteria lysate at 29,000 g

Peer J

for 1 h. Clarified lysates were incubated with 1 mg/mL protamine sulphate for 20 min at 4 °C to precipitate nucleic acid contaminants, which were removed by re-centrifugation at 29,000 g for 20 min. The lysate was loaded onto TALON metal affinity resin (Clontech, Mountain View, California, USA) in a gravity flow column. After washing with 25 volumes of lysis buffer, 25 volumes of lysis buffer containing 5 mM imidazole, and finally a further 25 volumes of lysis buffer containing 10 mM imidazole, the MNV NS6 proteins were eluted with a step gradient to 100 mM imidazole. The $His_6$ tag was removed during overnight dialysis in lysis buffer in the presence 2 mM $CaCl_2$, and 10 units of thrombin (Sigma-Aldrich, St. Louis, Missouri, USA) per mg of MNV NS6 protein. Following dialysis, uncleaved material and cleaved tags were removed in a second round of TALON purification. As a final polishing step, the unbound fractions containing cleaved MNV NS6 variants were pooled, concentrated and loaded onto a Superdex-75 size-exclusion column (mounted on AKTA FPLC; GE Healthcare, Little Chalfont, Buckinghamshire, UK) equilibrated with 50 mM Tris pH 8.0, 100 mM NaCl, 1 mM DTT (SEC buffer).

In the case of NS6 4′ and NS6 4′ 2|3 constructs the above protocol (and additives) were used but the lysis buffer was composed instead of 50 mM HEPES pH 6.5, 300 mM NaCl, 1 mM DTT. Size-exclusion chromatography was performed in 25 mM Tris pH 8.0, 200 mM NaCl, 5 mM DTT.

## Crystallisation of MNV NS6 variants

All crystallisation experiments were performed with a protein concentrations of 10–14.5 mg/mL in the SEC buffer used for the final purification step. Efforts to use the original crystallisation conditions to reproduce the MNV NS6 protein crystals with the packing arrangement that would place the C-terminus of one molecule in the active site of a neighbour (*Leen, Baeza & Curry, 2012*) failed for the four extended constructs generated in this study. Seeding equilibrated drops with fragments from extant NS6 crystals generated very small crystals that appeared similar in habit to the original crystals of MNV NS6 but they did not diffract appreciably.

We therefore widened the search for crystallisation conditions, carrying out screening on a sub-microlitre scale by sitting-drop vapour-diffusion. A 100 nL amount of protein solution at 14 mg/mL was mixed with 100 nL reservoir solution and equilibrated against 90 µL of reservoir solution using a Mosquito automated pipetting system (TTP LabTech, Cambridge, UK) and the following commercial crystallisation screens: Wizard 1 & 2 (Emerald Biosystems, Bainbridge Island, Washington, USA), PACT Premier, JCSG+, Morpheus and PGAScreen (Molecular Dimensions, Altamonte Springs, Florida, USA). Larger crystals were grown from initial hits by sitting-drop vapour diffusion at 18 °C in CompactClover Plates EBS-XPT (Jena Bioscience, Jena, Germany).

Useable crystals of MNV NS6 1′ were obtained by mixing 1 µL of protein with 1 µL mother liquor consisting of 10% (v/v) poly-ethylene glycol (PEG) 10000, 20% (v/v) ethylene glycol, 0.1 M MES/Imidazole pH 6.3. Crystals with cuboid shape appeared between 2 and 5 days and grew to full size (80 × 50 × 50 µm) in about 15 days and were flash-cooled in liquid nitrogen prior to data collection without any additional cryo-protectant.

Crystals of MNV NS6 2′ were obtained in 15% (v/v) PEG 3350, 0.1 M glycine, 0.1 M Na-citrate pH 7.0. They grew as long rods in 1–2 days and achieved full size (400 × 30 × 30 μm) in 5–10 days. The crystals were cryo-protected in mother liquor supplemented with 15% (v/v) PEG 200 and flash-cooled in liquid nitrogen.

Crystals of NS6 4′ and NS6 4′ 2|3 grew from 0.2 M KSCN, 0.1 M Bis-Tris propane pH 6.5–7.5, 20% w/v PEG 3350. They were cryo-protected adjusting the mother liquor solution to a final concentration of 30% (v/v) PEG 3350.

## X-ray data collection and processing

X-ray diffraction data from crystals of NS6 1′ and NS6 2′ were collected on a Pilatus 6M-F detector at the I03 beamline at the Diamond Light Source (Didcot, UK). For the MNV NS6 1′ crystals a 2.3 Å data set of 200 frames was collected with an oscillation width of 1° per frame. Diffraction images were integrated and scaled using the CCP4 program suite (*Winn et al., 2011*). Data-collection statistics are summarized in Table 1.

For the MNV NS6 2′ crystals, 720 frames with a 0.5° oscillation were collected. A 3.1 Å dataset was integrated and scaled as described above; see Table 1 for data collection statistics.

MNV NS6 4′ and NS6 4′ 2|3 crystals data was collected in-house using a Rigaku MicroMax-007 HF-M X-ray generator and a Saturn 944+ CCD detector. Data sets of 316 and 344 0.5° oscillation frames were collected for NS6 4′ and NS6 4′ 2|3 crystals at 2.47 and 2.42 Å respectively. The data were processed and scaled as described above.

All four datasets were submitted to the UCLA MBI Diffraction Anisotropic Server (http://services.mbi.ucla.edu/anisoscale) for anisotropic analysis. Following the server indication of severe anisotropy diffraction of the MNV NS6 1′ and NS6 2′ crystals both data sets were truncated/scaled using the server default values, in particular using a 3.0 cut-off for F/sigma (*Strong et al., 2006*; *Sawaya, 2014*).

## Phasing, model building and refinement

Molecular-replacement phasing was performed in Phaser (*McCoy et al., 2007*) using the crystal structure of full-length MNV NS6 (PDB entry 4ASH) (*Leen, Baeza & Curry, 2012*) pruned of double conformations and the initial 6 and terminal 11 residues to avoid biasing the conformations of the termini.

The MNV NS6 1′ and NS6 2′ structures obtained from molecular replacement were subjected to restrained refinement using REFMAC (*Murshudov et al., 2011*), in the CCP4 program suite (*Collaborative Computational Project , 1994*). Molecular replacement solutions of NS6 4′ and NS6 4′ 2|3 were refined using Phenix refine (*Adams et al., 2010*). All manual model adjustments were made in Coot (*Emsley & Cowtan, 2004*).

## RESULTS AND DISCUSSION

### Structure determination

Datasets collected for MNV NS6 1′ and NS6 2′ were initially scaled and truncated—at 2.3 Å and 3.1 Å respectively—with the assumption in each case that the diffraction was isotropic even though some anisotropy was evident in the diffraction images. This approach resulted

**Table 1** Data-collection and model refinement statistics.

| | MNV NS6 1′ | MNV NS6 2′ | MNV NS6 4′ | MNV NS6 4′ 2\|3 |
|---|---|---|---|---|
| **Data collection** | | | | |
| Radiation source | Diamond I03 | Diamond I03 | Rigaku MicroMax-007 HF-M | Rigaku MicroMax-007 HF-M |
| Wavelength (Å) | 1.000 | 1.000 | 1.54 | 1.54 |
| Detector | Pilatus 6M-F | Pilatus 6M-F | Saturn 944+ CCD | Saturn 944+ CCD |
| Resolution limits[a](Å) | 70.93–2.3 (2.42–2.3) | 68.02–3.1 (3.31–3.1) | 38.11–2.472 (2.561–2.472) | 19.27–2.417 (2.503–2.417) |
| Space group | C2 | P6$_1$22 | C2 | P1 |
| Unit-cell parameters (Å,°) | $a = 99.64$ $b = 111.86$ $c = 81.29$ $\beta = 119.24$ | $a = 136.04$ $c = 82.39$ | $a = 88.19$ $b = 35.36$ $c = 52.81$ $\beta = 105.98$ | $a = 35.52$ $b = 47.32$ $c = 53.07$ $\alpha = 104.45$ $\beta = 91.53$ $\gamma = 110.61$ |
| Mosaicity (°) | 0.42 | 0.70 | 0.67 | 1.11 |
| Number of unique reflection | 34,593 | 8,555 | 5,711 | 10,817 |
| Multiplicity | 3.5 (3.6) | 5.7 (6.0) | 3.0 (2.3) | 1.8 (1.8) |
| $\langle I/\sigma\,(I)\rangle$[b] | 6.7 (1.2) | 10.9 (2.1) | 19.63 (6.94) | 6.7 (2.31) |
| Completeness (%) | 99.7 (99.8) | 99.8 (99.6) | 99.03 (92.44) | 91.14 (66.47) |
| $R_{merge}$ (%)[c] | 8.4 (8.6) | 8.5 (91.5) | 3.4 (10.6) | 9.3 (27.1) |
| Overall $B$ factor (Å$^2$) | 54.4 | 94.3 | 27.1 | 24.9 |
| **Model refinement** | | | | |
| Number of non-hydrogen atoms/waters | 5,356/93 | 2,563/0 | 1,218/22 | 2,562/107 |
| $R_{work}$ (%)[d] | 23.0 | 25.4 | 20.9 (27.2) | 21.0 (28.3) |
| $R_{free}$ (%)[e] | 27.8 | 30.3 | 25.2 (37.1) | 26.2 (36.8) |
| RMSD bonds (Å)[f] | 0.012 | 0.012 | 0.009 | 0.003 |
| RMSD bond angles (°) | 1.605 | 1.610 | 1.08 | 0.67 |
| Ramachandran plot (% favoured/allowed) | 89.3/10.7 | 88.8/11.2 | 97.5/2.5 | 96.1/3.9 |
| PDB identifier | 4x2v | 4x2w | 4x2x | 4x2y |

**Notes.**

[a] Values in parentheses refer to the highest resolution shell of data.

[b] $\langle I/\sigma\,(I)\rangle$ is the mean signal-to-noise ratio, where $I$ is the integrated intensity of a measured reflection and $\sigma\,(I)$ is the estimated error in the measurement.

[c] $R_{merge} = 100 \times \Sigma_{hkl}|I_j(hkl) - \langle I_j(hkl)\rangle|/\Sigma_{hkl}\Sigma_j I(hkl)$, where $I_j(hkl)$ and $\langle I_j(hkl)\rangle$ are the intensity of measurement j and the mean intensity for the reflection with indices hkl, respectively.

[d] $R_{work} = 100 \times \Sigma_{hkl}\|F_{obs}| - |F_{calc}\|/\Sigma_{hkl}|F_{obs}|$.

[e] $R_{free}$ is the $R_{work}$ calculated using a randomly selected 5% sample of reflection data that were omitted from the refinement.

[f] RMSD, root-mean-squared deviations (from ideality).

in data sets with relatively high values of $R_{merge}$ and low signal-to-noise ratios [$I/\sigma\,(I)$] for the highest-resolution shells of data (Table 1). The MNV NS6 1′ crystals were determined to be composed of four molecules in the asymmetric unit of the C2 unit cell, which is consistent with a solvent content of 50.3% (*Matthews, 1968*). MNV NS6 2′ crystals belong to space-group P6$_1$22 and were determined to contain two molecules in the asymmetric unit, with slightly higher solvent content of 55.3%.

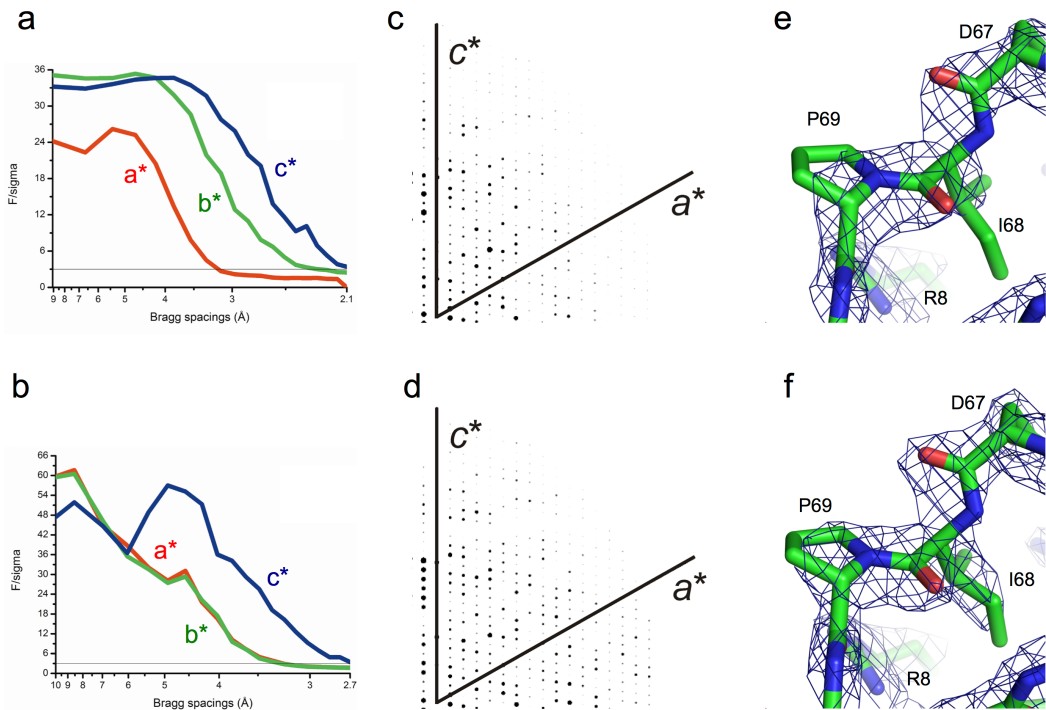

**Figure 2 Analysis and correction of the anisotropic diffraction observed for crystals of NS6 1′ and NS6 2′.** F/sigma versus Bragg spacings for each of the cell directions for (A) NS6 1′ and (B) NS6 2′ respectively. Pseudo-precession images of the anisotropy in the a\*c\* (h0l) plane for NS6 1′ (C) before and (D) after correction. 2$F_o$-$F_c$ electron density maps contoured at 2$\sigma$ after one round of refinement of the molecular replacement solutions obtained with *Phaser* (*McCoy et al., 2007*) for NS6 1′ (E) before and (F) after anisotropic correction.

Initial structural factors (processed isotropically) were used for molecular replacement, and in the case of MNV NS6 1′ produced four possible solutions with a highest LLG of 2,723 and TFZ of 19.7. However, initial refinement of the model obtained by molecular replacement stalled at relatively high values of $R_{work}$ (∼33%) and $R_{free}$ (∼39%). A similar problem was encountered with initial attempts to refine the MNV NS6 2′ model.

At this stage we re-visited the processed data to determine if the anisotropy was at the root of the refinement problems. Plots of F/sigma against resolution for each of the 3 principal axes revealed the severity of the anisotropy of the diffraction from both crystals (Figs. 2A, 2B); when truncated at F/sigma <3.0 along each axis, the spread of data was ellipsoidal in appearance. The server identified the c\* axis as stronger diffracting than the a\* and b\* directions in each case, and detected anisotropic $\Delta B$ values of 68.6 Å$^2$ and 51.0 Å$^2$ for MNV NS6 1′ and NS6 2′, respectively. Anisotropic $\Delta B$ reports the directionality dependence of the intensity falloff with resolution (http://services.mbi.ucla.edu/anisoscale/). Following the F/sigma analysis, the MNV NS6 1′ data were truncated to 3.1 Å, 2.3 Å, and 2.1 Å along a\*, b\* and c\*, respectively. To create a nominally isotropic data set, B-factor corrections of 39.2, −9.8 and −29.4 Å$^2$ were applied to the observed structure factors along the a\*, b\* and c\* directions respectively. The reduction in the anisotropy of the corrected data in the a\*c\* (h0l) plane can been seen by comparing Figs. 2C and 2D.

Following anisotropy analysis the MNV NS6 2′ data were truncated to 3.2 Å, 3.2 Å, and 2.7 Å along $a*$, $b*$ and $c*$ directions. To generate the nominally isotropic data set, $B$-factor correction of 17.0, 17.0 and −34.0 Å$^2$ along the same axes were applied to the observed structure factors. The anisotropic truncation of the data, with the new limit of 2.7 Å along the $c*$ axis, resulted in a dramatic increase in the number of unique reflections from 8,555 in the "pre-treated" data to 12,786.

MNV NS6 4′ or NS6 4′ 2|3 data sets plots of F/sigma against resolution for each of the 3 principal axes revealed only mild anisotropy of the diffraction from both crystals (anisotropic $\Delta B$ values of 13.3 Å$^2$ and 15.4 Å$^2$ for MNV NS6 4′ and NS6 4′ 2|3, respectively). In these cases, no directional-dependent truncation of the data was applied.

After taking account of the anisotropy of the data, the axes-dependent truncated and corrected structure factors were then used to repeat the molecular replacement phasing of the NS6 1′ data and resulted in significantly better solutions. The best solution had an LLG of 3051 and TFZ of 32.3. Moreover, the model showed immediate improvement in the early cycles of refinement ($R_{work}$ ∼31%; $R_{free}$ ∼36%) and yielded electron density maps that were much more interpretable (Figs 2E and 2F). Multiple cycles of refinement and model building lead to a structure characterized by $R_{work}$ of 23.0% and $R_{free}$ of 27.8%.

Anisotropic treatment of the MNV NS6 2′ data also improved the molecular replacement outcome, with a unique solution found by Phaser in contrast with the two possible solutions of the uncorrected data. The LLG value increased markedly from 385 to 887 while there was a slight drop in the value of TFZ (from 19.2 to 18.8). Nevertheless, this resulted in immediate improvements in refinement: $R_{work}$ dropped from 32.5 to 30.3% while $R_{free}$ was reduced from 42.0 to 39.0%. These statistics were further improved to $R_{work}$ of 25.4% and $R_{free}$ of 30.3% in the final structure.

The translation Z-scores and LLG's for the Phaser molecular replacement solutions for MNV NS6 4′ are 13.8 and 1,042, respectively, and for MNV NS6 4′ 2|3 27.1 and 1,394, respectively. The crystals structures of NS6 4′ and NS6 4′ 2|3 were refined to $R$ factors ($R_{work}/R_{free}$) of 20.9/25.2 and 21.0/26.2%, respectively.

### Structure analysis

Overall, the structure of the protease core domains for NS6 1′, NS6 2′, NS6 4′ and NS6 4′ 2|3 is very similar to that reported previously for MNV NS6 (*Leen, Baeza & Curry, 2012*)—(root mean squared differences are less than 1 Å) (Fig. 3). For three of the four new structures (NS6 2′, NS6 4′ and NS6 4′ 2|3) it was possible to identify conserved crystal contacts, involving a two-fold symmetric packing arrangement largely mediated by the short N-terminal helix and the loop connecting $\beta$-strand aII and bII (Fig. 3B). In each case the C-terminus of at least one protein molecule in the asymmetric unit was observed to extend away from the body of the protease and to interact with a neighbouring molecule in the crystal—as had been observed previously for MNV NS6. Moreover, the extended C-terminus invariably inserted into the peptide-binding groove formed between loops bII-cII and eII-fII loops (Fig. 4). Disappointingly, however, in no case did the extended peptide make interactions that were consistent with specific contacts with the protease

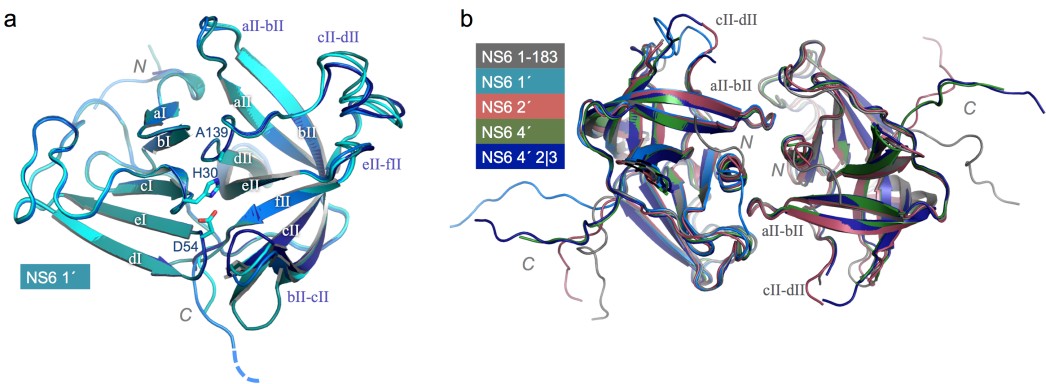

**Figure 3 Structure and conserved packing interfaces of C-terminally extended NS6 proteases.** (A) Superposition of the four molecules in the asymmetric unit of crystals of NS6 1′ (coloured various shades of blue). The β-strands are labelled, as are the N- and C-termini and the conserved side-chains of catalytic triad. Note that the active site C139 has been replaced by A139 in all structures reported here. (B) A conserved packing arrangement observed for the mature NS6 protease (NS6 1–183) (*Leen, Baeza & Curry, 2012*) and three of the structures solved in the present work (NS6 2′, NS6 4′ and NS6 4′ 2|3). In this panel the label for the N-terminus is adjacent to the N-terminal helix that is central to the packing interface. This packing arrangement is not conserved in the NS6 1′ crystals but the B chain of NS6 1′ is included, superposed on the molecule on the left-hand side, to further illustrate the variation in the C-termini of the different constructs. Key features are labelled to facilitate comparison of (A) and (B).

active site. We will briefly describe each of the four new structures before turning to the question of why in no case a protease–substrate complex was obtained.

MNV NS6 1′ crystallised in space group *C2* with four molecules in the asymmetric unit. This structure is notable for the fact that, in contrast to the other three structures reported here and the structures previously reported for Norwalk virus NS6 protease (*Zeitler, Estes & Prasad, 2006*; *Muhaxhiri et al., 2013*), the electron density map was of sufficient quality in all four molecules of the asymmetric unit to permit the incorporation of the loop between the β-strands cII and dII (Fig. 3A). Superposition of the four copies of NS6 1′ indicates some structural variation in this loop—the $C_\alpha$ positions vary by 1–2 Å—consistent with the notion that it is rather flexible (Fig. 3A).

In three of the molecules in the asymmetric unit the C-terminus is disordered beyond residue 173 (chains A and D) or residue 174 (chain C). The electron density for the last four residues modelled in chain D (Ala 170 to Gly 173) is poor, presumably due to disorder. Nevertheless, we included these residues in the final model because removal increased the *R*-factor.

Although the full C-termini of chains A, C and D of the NS6 1′ could not be modelled, the electron density map revealed additional features that could be built as short stretches of polypeptide. These correspond to portions of the missing C-termini but, because of discontinuities in the electron density, it is not possible to unambiguously ascribe which monomer they belong to. These short segments were present in the asymmetric unit, and were modelled as (i) a stretch of five residues (Chain E) that could be assigned confidently as corresponding to residues Leu 180 to Gly 184 that lies near to the active site of a neighbouring chain A; and (ii) an extended portion of electron density that

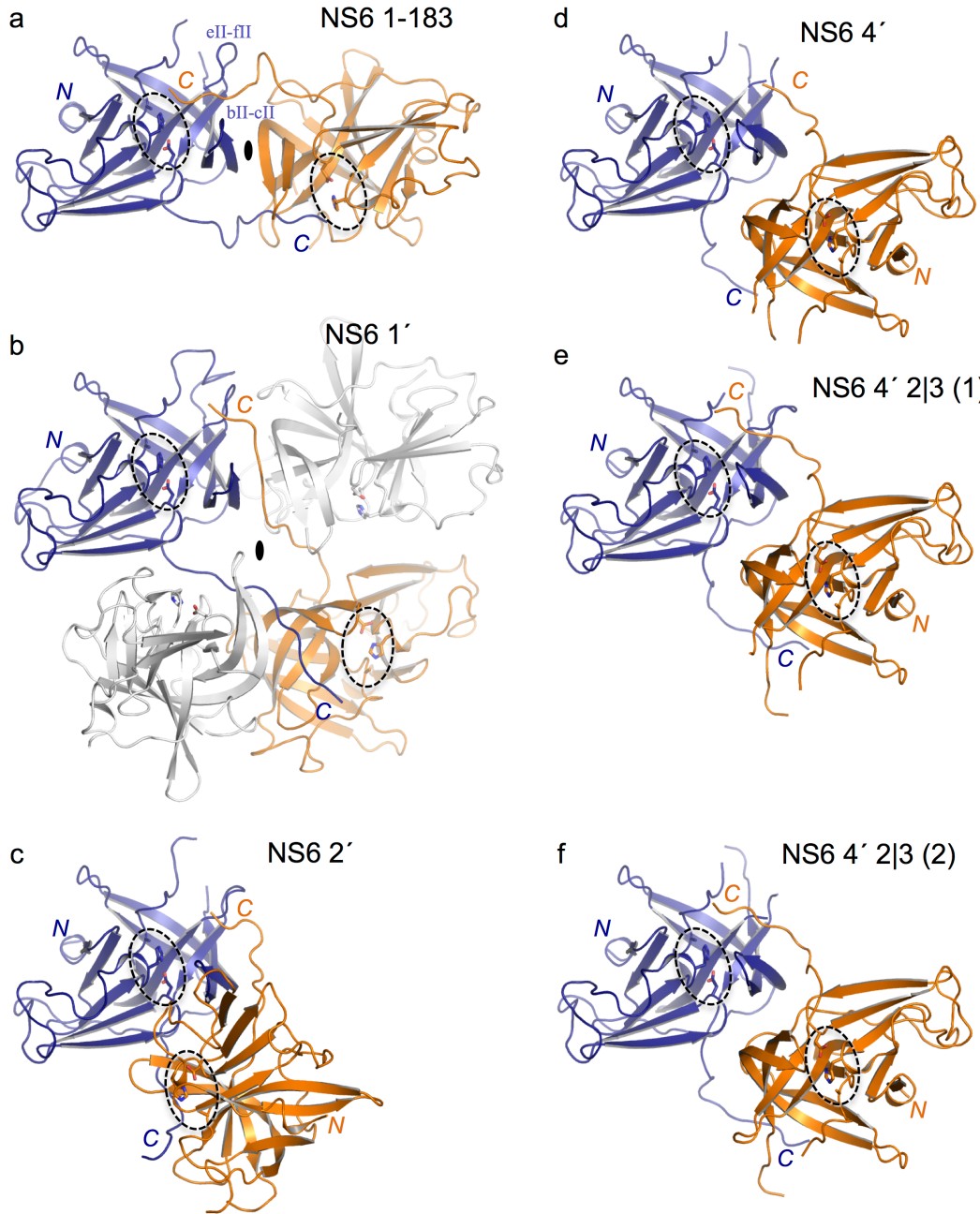

**Figure 4 Comparison of the packing arrangements of C-terminally extended NS6 proteases.** (A)The mature NS6 protease (NS6 1-183) (*Leen, Baeza & Curry, 2012*). Chains which interact via C-termini are coloured blue and orange. This colour-scheme is maintained throughout the figure; note also that the orientation of the blue chain is the same in each panel. In all panels the side-chains of the active site residues/mutations C139A, H30 and D54 are shown as sticks; their locations are indicated by dotted ovals. The presence of two-fold symmetry axes between interacting molecules are indicated by a solid black oval, although the views shown are only approximately along these axes. (B) NS6 1′—the interaction is between a pair of symmetry-related B chains. An additional pair of symmetry-related C chains, which also contact the extended C-terminus of the B chains, is shown in white (continued on next page...)

**Figure 4 (...continued)**

(C) NS6 2′. In this case the interaction is between the two chains in the asymmetric unit. (D) NS6 4′ —here there is only one chain in the asymmetric unit and the interaction is not symmetric. (E) NS6 4′ 2|3 —here again the interaction between chain B (blue) of one asymmetric unit and chain A (orange) of another is not symmetric. (F) NS6 4′ 2|3—a second but very similar mode of interaction in these crystals between chain A (blue) of one asymmetric unit and chain B (orange) of another.

was not of sufficient quality to identify the amino acid side chains and was built as a seven-residue poly-Ala peptide (chain F), that lies between chains A and D and may be part of a symmetry-related molecule of chain A. However, in neither case were specific interactions with the putative active sites of neighbour monomers observed.

In contrast to chains A, C and D, the electron density for the B chain in the asymmetric unit was sufficiently clear to permit inclusion of all 183 residues of NS6 and the extra first residue of NS7 (here labelled Gly 184) in the refined model (Fig. 4B). The C-terminus of Chain B of NS6 1′ beyond Gly 173 extends to make contacts with chain B from an adjacent asymmetric unit to which it is related by a two-fold symmetry axis. This two-fold symmetric packing arrangement is reminiscent of the inter-protein contacts previously observed for the full length NS6 (*Leen, Baeza & Curry, 2012*) (Fig. 4A). However, although the C-terminus of NS6 1′ comes close to the substrate-binding site of neighbouring proteases in the crystal, it does not reach far enough and makes none of the specific contacts needed to position the scissile bond (Gln 183-Gly 184) in the active site.

MNV NS6 2′ crystallised in space group $P6_122$ with two molecules in the asymmetric unit. The electron density indicates the positions of residues 3-181 in chain A and 4-181 in chain B, but is not good enough to allow complete modelling of the N- or C-termini. The two monomers found in the asymmetric unit are related by a quasi-2-fold symmetry axis and form an apparent homodimer with an interface area of 1,097 Å$^2$ (Fig. 4C). In addition to neighbourly contacts made by the two C-termini, dimerization within the crystal is stabilised by hydrogen bonds involving Ser 51, Ser 111 and Val 113 of chain A and Ser 51, Ser 52, Ser 111 and Val 113 of chain B. Although the C-termini within the asymmetric unit each embrace the other monomer, reaching into the groove formed between the bII-cII and eII-fII loops, they are again not located within the substrate-binding site.

MNV NS6 4′ crystallised in space group *C*2 but in a unit cell that is very different from the *C*2 crystals obtained with MNV NS6 1′ (Table 1) and that has only one molecule in the asymmetric unit. The electron density was sufficient to build a model that starts at residue 4 and ends at residue 179. Although the extended C-terminus of NS6 once again reaches between loops bII-cII and eII-fII of its near neighbour, the relation between the two molecules does not involve two-fold or quasi two-fold symmetry (Fig. 4D). However, no specific contacts are made by the C-terminal peptide with the substrate binding site.

NS6 4′ 2|3 crystallised with two molecules in a *P*1 unit cell. The modelled monomers start at residue 4 and end at residue 182; in contrast to the other structures reported here, the A chain lack density for the tip of the eII-fII loop so residues 162–163 were omitted. The A and B chains each extend their C-termini into the cleft formed between the bII-cII and eII-fII loops of a neighbouring molecule (Figs. 4E and 4F). The packing arrangements

are distinct but rather similar and, once again, lack specific contacts that are consistent with productive substrate binding.

## Implications of the structures

With the extended NS6 protease constructs designed for this study we aimed to exploit our earlier finding that the mature protease crystallised with the C-terminus of one molecule inserted into peptide binding site of a neighbouring protein in a way that revealed specific protease-product interactions (*Leen, Baeza & Curry, 2012*). We had hoped to obtain the same packing interaction with the C-terminally extended complexes in order to determine the structures or protease–substrate complexes that would reveal the details of the interactions made by the amino acids in positions P1′-P4′ of the substrate, but in each of the four cases that we probed, extension of the C-terminus resulted in novel packing arrangements, none of which captured a proper protease–substrate reaction.

Why did we not get the same crystal form as for the full-length protein? At present it is difficult to give a definitive answer to this question. Although we screened extensively for crystallisation conditions, we cannot claim to have searched exhaustively and it may be that further efforts might yet succeed. Moreover, it is well known that even very modest changes to protein constructs may alter substantially the way that they crystallise, though it is worth noting that a similar strategy to explore the specific interactions of different protease-product complexes was applied successfully to Norwalk virus NS6 (*Muhaxhiri et al., 2013*). In preliminary investigations using nuclear magnetic resonance (NMR) to investigate solution structures of extended MNV NS6 protease structures, we have obtained evidence to suggest that their C-termini may have a propensity to fold into the active site of the molecule that it belongs to, making a *cis* interaction (H Fernandes, 2015, unpublished data). This may explain why *trans* interactions were not observed in our crystals but in turn raises a further question: why was this conformational state not captured in the crystal form? One possibility is that the concentration of proteins inevitably involved in crystallisation somehow destabilizes the *cis* interaction but does not necessarily capture a catalytically competent *trans* interaction between the cleavage junction within the extended C-terminus and a protease neighbour. Why this would be the case, particularly when a protease–substrate complex would be expected to be a relatively stable state, remains a mystery.

## ACKNOWLEDGEMENTS

We thank staff at the Diamond Light Source (Didcot, UK) for assistance with data collection.

### Funding

This work was supported in part by grant funding from the Biotechnology and Biological Sciences Research Council, UK (Ref: BB/J001708/1). The funders had no role in study design, data collection and analysis, decision to publish, or preparation of the manuscript.

### Grant Disclosures

The following grant information was disclosed by the authors:
Biotechnology and Biological Sciences Research Council, UK: BB/J001708/1.

### Competing Interests

The authors declare there are no competing interests.

### Author Contributions

- Humberto Fernandes and Eoin N. Leen conceived and designed the experiments, performed the experiments, analyzed the data, contributed reagents/materials/analysis tools, wrote the paper, prepared figures and/or tables, reviewed drafts of the paper.
- Hamlet Cromwell Jr and Marc-Philipp Pfeil performed the experiments, analyzed the data, contributed reagents/materials/analysis tools, reviewed drafts of the paper.
- Stephen Curry conceived and designed the experiments, analyzed the data, contributed reagents/materials/analysis tools, wrote the paper, prepared figures and/or tables, reviewed drafts of the paper.

### Data Deposition

The following information was supplied regarding the deposition of related data:
RCSB protein data bank:

http://www.rcsb.org/pdb/explore/explore.do?structureId=4x2v
http://www.rcsb.org/pdb/explore/explore.do?structureId=4x2w
http://www.rcsb.org/pdb/explore/explore.do?structureId=4x2x
http://www.rcsb.org/pdb/explore/explore.do?structureId=4x2y

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
