# Peer review of "Structure determination of Murine Norovirus NS6 proteases with C-terminal extensions designed to probe protease–substrate interactions"

_PeerJ, doi:10.7717/peerj.798_

## Round 0.1 · original submission · Major Revisions

Would you like to answer the reviewers questions and try to correct your manuscript according to reviewers criticism.

Reviewer 1 ·

Basic reporting

No comments.

Experimental design

This reviewer thinks that a conformation of the protein in crystals is often artificial, which means that crystals are very much affected by chance. The authors would be lucky when they got a crystal described in their previous paper (PLoS One (2012, 6, e38723). This reviewer is wondering if the authors should try to co-crystallize with a peptide substrate.

Validity of the findings

There is nothing up-to-date, but this reviewer understands the authors' effort.

·

Basic reporting

This paper is written in a very clear, concise, style - although the Introduction gives an excellent general background and description of previous research ffindings that lead to the research presented in this paper. Similarly, the data are very clearly presented and the correct interpretations drawn. The Materials and Methods section provides sufficient detail for the work to be reproduced. Similarly, the 'Results and Discussion' section provides enough detail as to the methodology of the structure determination, structure analysis and is followed by an insightful discussion of the (biological) implications of the reported structures. A clear, logical, structure to the paper which was very straightforward to read and comprehend - even for someone not directly involved in X-ray crystalogrphic work.

Experimental design

I have no criticisms of the experimental design, the quality of the data nor the interpreatations placed on the data.

Validity of the findings

No comments - a high quality paperwritten to highest of standards.

Comments for the author

It may help the (non-proteinase) reader what - specifically - is meant by the term 'mature'.

Reviewer 3 ·

Basic reporting

Noroviruses are positive-sense single-stranded RNA viruses which are responsible for over half of the outbreaks of gastroenteritis worldwide. Any studies related to the viruses are important and required for public health. After entering the cells, like other +strand RNA viruses, such as astroviruses, picornaviruses and coronaviruses, the virus translate a polyprotein that is then cleaved by the virally encoded protease to release various non-structural viral proteins required for its replication. In noroviuses, NS6 protease located within the C-terminal half of the polyprotein performs all the processing, including its own autocleavage.
In recent years, the authors’ lab and others have determined the NS6 proteases from several norovirus strains either by X-ray crystallography or NMR. NS6 is a cysteine protease with a chymotrypsin-like fold: two β-barrel domains separated by a cleft that contains a Cys-His-Asp/Glu catalytic triad. Adventitious crystal contacts placed the C-terminus of one molecule in the active site of another, thereby generating a structure of a protease-product complex that can represent the final step of the “trans” cleavage and the specific contacts made by the P5-P1 residues of the product. In the present studies, they want to use similar crystal packing contacts to find out the possible binding mode for P1’-P4’ residue of the substrate but unfortunately, failed.
Major concerns:
1. In the cases of NS6 C139A 1’ and 2’, the authors tried to correct their severe anisotropy diffraction by UCLA MBI Diffraction Anisotropic Sever and solved the two structures. Undoubtedly, they used the right tools and ask the right question. However, they should try more to solve the problem, not just stop here. At least, they can try more inactive mutants’ crystallization to achieve the goal, such as C139S (active or inactive?), D54A, D54N…etc. It is still too early to give up.
2. What and how is the “details of the specific contacts made by the P5-P1 residues”? The authors should give us more information about the binding and S5-S1 subsites, not just ask us to find it out from their previous work. Furthermore, although “fail”; I am still very interested to take a look the varied C-termini location more detail, not just a few “lines”. At least, give us a predicted functional active P1’-P4’ sites and compare the difference more detail.
3. Confusing figures: in figure 2, I cannot see the NS6 1’ in fig 2b. At least they should put B chain of NS6 1’ in and make a comparison. By the way, some important regions such as the N terminal short helix and αII-βII loop, should be labeled in fig 2b. In fig 2a, the modeled P5-P1 residues from other structures should be put in. In fig. 3, 2-fold axis should be labeled in some subfigures, if have. By the way, where is the catalytic triad in fig. 3’s structure? Show it.

Minor concerns
1. Crystallize or crystallise? Choose one and let it all the same.

Experimental design

See basic reporting.

Validity of the findings

See basic reporting.

Comments for the author

See basic reporting.

---

## Round 0.2 · accepted · Accept

I think your manuscript is ready for publication.